# QPT V2: Masked Image Modeling Advances Visual Scoring

## ABSTRACT

Quality assessment and aesthetics assessment aim to evaluate the perceived quality and aesthetics of visual content. Current learning-based methods suffer greatly from the scarcity of labeled data and usually perform sub-optimally in terms of generalization. Although masked image modeling (MIM) has achieved noteworthy advancements across various high-level tasks (*e.g.*, classification, detection *etc.*). In this work, we take on a novel perspective to investigate its capabilities in terms of *quality- and aesthetics-awareness*. To this end, we propose **Q**uality- and aesthectics-aware **P**re**T**raining (QPT V2), the first pretraining framework based on MIM that offers a unified solution to quality and asthetics assessment. To perceive the high-level semantics and fine-grained details, pretraining **data** is curated. To comprehensively encompass quality- and aesthetics-related factors, **degradation** is introduced. To capture multi-scale quality and aesthetic information, **model** structure is modified. Extensive experimental results on 11 downstream benchmarks clearly show the superior performance of QPT V2 in comparison with current state-of-the-art approaches and other pretraining paradigms.

## CCS CONCEPTS

• **Computing methodologies** → **Artificial intelligence**; **Computer vision**; **Computer vision tasks**; **Scene understanding**;

## KEYWORDS

visual scoring, quality and aesthetics assessment, self-supervised learning, masked image modeling

## 1 INTRODUCTION

The aims of Image Quality Assessment (IQA), Visual Quality Assessment (VQA), and Image Aesthetics Assessment (IAA) are to appraise the quality and aesthetics of visual content, serving as critical components across a multitude of vision applications including video enhancement, transcoding, and transmission [34, 65, 105]. While being studied separately for a considerable period, these tasks present strong resemblance in various aspects. *All these tasks share the same core objective, that is, to mimic the Human Visual System (HVS), so as to generate accurate scores aligned with human perception* [27, 82, 83]. Moreover, the proliferation of User-Generated Content (UGC) [73, 77, 110] and AI-Generated Content (AIGC) [5, 93, 102, 103] has become a trend in recent years, which greatly contributed to the exponential growth of image and video data [3].

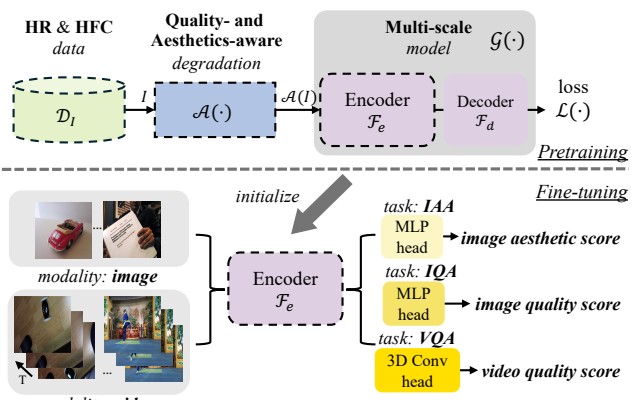

**Figure 1: QPT V2: a new MIM-based pretraining paradigm for visual scoring. For pretraining, dataset $\mathcal{D}_I$ provides HR & HFC images, augmented by quality- and aesthetics-aware degradation $\mathcal{A}(\cdot)$. A multi-scale autoencoder $\mathcal{G}(\cdot)$ outputs the reconstructed images. Through finetuning of the encoder, it can solve visual scoring tasks like IQA, VQA, and IAA.**

The complexity and interrelation of quality-related and aesthetics-related factors in emerging content are unprecedented, and analyzing single factors alone is insufficient to achieve a comprehensive perception of visual content aligned with human perception. In response to the aforementioned resemblance and trend, we refer to IQA, VQA, and IAA jointly as Visual Scoring (VS) for analysis.

Facilitated by the advancements of deep neural networks [13, 26, 41, 70, 81], learning-based methods [28, 43, 44, 72, 112] have surpassed traditional methods [10, 57, 78, 92] based on handcrafted features on multiple VS benchmarks [6, 9, 17, 20, 59, 86, 101]. They acquire features with strong expressiveness via regressing from the Mean Opinion Scores (MOS). However, one of the primary obstacles in solving VS lies in the limited size of labeled datasets [44, 46, 55, 74, 111]. Due to the high cost associated with collecting MOS through extensively annotated subjective studies, the scale of VS datasets is often only a fraction, ranging from one-tenth to even one-hundredth, of other high-level visual task datasets (*e.g.*, object recognition). At all events, the paucity of labeled data restricts the capabilities of data-driven deep learning methods.

To tackle this problem, some previous efforts increased data size by patch/frame-level augmentation [4, 35, 36, 46] or mixed-database training [39, 40, 94, 107]. However, the quality and aesthetics scores of local patches often differ from those of the entire content, and subjective differences are observed across datasets, thus hindering the achievement of promising results. On the other hand, a different research line [37, 76, 83, 89, 95] exploits knowledge valuable for VS from **datasets** and **model weights** of other domains, by tapping into the power of pretrained vision or vision-language (VL) models [63, 76, 89, 95, 98, 100, 106]. These works attempt to extract knowledge that is more quality- or aesthetic-aware from large-scale datasets by carefully designing **pretraining objectives**[44, 55], and

are then finetuned on downstream VS tasks. The pretraining objectives of existing works are mainly based on contrastive learning [67, 80], which can be viewed as a global self-supervised learning (SSL) approach, as it groups similar samples closer and diverse samples far from each other [25, 55, 111]. However, this "sample-level" discernment is insufficient for capturing local distortions and visual attributes [60]. Therefore, exploring more effective pixel-level discrimination may be beneficial for incorporating pretrained priors into downstream VS tasks.

Masked Image Modeling (MIM) [24], which learns representation by pixel-level reconstruction of the masked regions in the input, has demonstrated its impressive ability of semantic- and texture-aware perception in visual tasks [2, 61, 84, 109]. In this paper, we conduct a detailed exploration of MIM, in which we observe MIM can learn both sample-level and pixel-level information of the visual content, showing the potential to serve as a general pretraining recipe to VS tasks. As shown in Fig. 1, we propose QPT V2, the first pretraining framework based on MIM that offers a unified solution for VS tasks. To enhance the acquisition of prior knowledge by MIM for VS tasks, we propose further improvements and optimizations of the vanilla MIM from the perspectives of **data**, **degradation**, and **model**. *Regarding the realm of data*, we curate a dataset with high resolution (HR) and high foreground coverage (HFC), thereby aiding the pretext task of MIM. *Regarding the realm of degradation*, we propose an optimal strategy for applying degradations to the reconstruction target, exploring the type and composition of degradation to acquire prior knowledge of practical scenarios. *Regarding the realm of model*, we use a drop-in strategy to learn multi-scale representations by adaptively fusing features of different layers. Our main contributions can be summed as follows:

- To the best of our knowledge, we are the first to validate the capability of MIM in adeptly unifying downstream visual scoring tasks. We decompose MIM into three crucial components: data, degradation, and model, and individually investigate their respective influences.
- We propose QPT V2, which stands as the pioneering MIM-based pretraining framework, offering a unified solution for VS tasks. To enhance the acquisition of prior knowledge through MIM, we make targeted improvements in the aspects of data, degradation, and model.
- QPT V2 achieves state-of-the-art (SOTA) results on 11 benchmarks in IQA, VQA, and IAA, surpassing other pretraining paradigms as well. Extensive ablation studies prove the validity of each enhancement of MIM.

## 2 RELATED WORK

### 2.1 Visual Scoring

Visual scoring necessitates precise scoring of visual content in terms of quality (*e.g.* , IQA, VQA) and aesthetics (*e.g.* , IAA). In this work, we focus on Non-reference QA (*e.g.* , NR-IQA and NR-VQA), since the availability of pristine data is too hard in the real world. At the early stage, handcrafted features based on natural scene statistics (NSS) dominate the realm of VS [56, 57, 69]. Later, data-driven methods enhanced the performance significantly with the rise of deep learning [15, 35, 45, 83, 89]. Nonetheless, they rely heavily on label-intensive supervision. Previous works attempt to

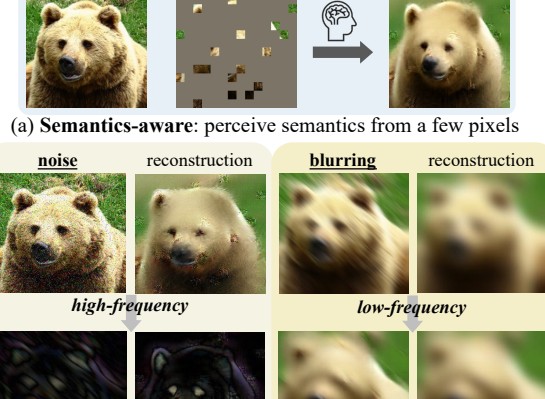

(a) **Semantics-aware**: perceive semantics from a few pixels

(b) **Distortion-aware**: perceive distortions by reconstruction

**Figure 2: Semantics- and distortion-awareness of the pixel-based MIM (a) MIM has the ability to understand the semantics; (b) Pixel-based MIM can reconstruct the distortions applied original images, the left column and the right column are high and low frequency intervals, respectively.**

solve this problem by data augmentation [4, 35, 36, 99], mixed-database training [40, 46, 74, 94, 107], rank-based learning [47, 50] and general knowledge transfer [15, 83, 89, 108].

Several researches focus on extracting quality or aesthetics information by large-scale pretraining. Among them, CONTRIQUE [55] learns distortion-related information on images with synthetic and realistic distortions based on contrastive learning. Similarly, Re-IQA [66] re-engineers the MoCo-v2 [8] framework and applies intricate data augmentations to learn quality-aware features. Moreover, QPT [111] introduces a diverse array of degradations and composites to mimic real-world distortions, which greatly expands the pretraining data volume. Different from them, we devise a pretraining framework based on MIM to learn effective quality- and aesthetics-related representations.

### 2.2 Masked Image Modeling

Masked modeling learns representation by reconstructing a masked portion of the input. Driven by the success of BERT [12] in NLP, MIM has become an representative SSL method in computer vision [2, 24, 61, 84, 109]. As a pioneer work, BEiT [2] proposes to reconstruct the features of DALL-E [64]. MAE [24] directly reconstructs raw pixels of the masked areas, which greatly simplifies the whole pretraining pipeline. Some studies prove that pixel-based MIM is biased towards reconstructing low-level details, thus hindering the performance on high-level tasks [2, 51, 60]. As a result, following works introduce more complicated reconstruction target rather than using raw pixels [16, 33, 87, 88]. While previous MIM studies mainly focus on high-level tasks, in this paper, we make the first attempt to adapt MIM to visual scoring.

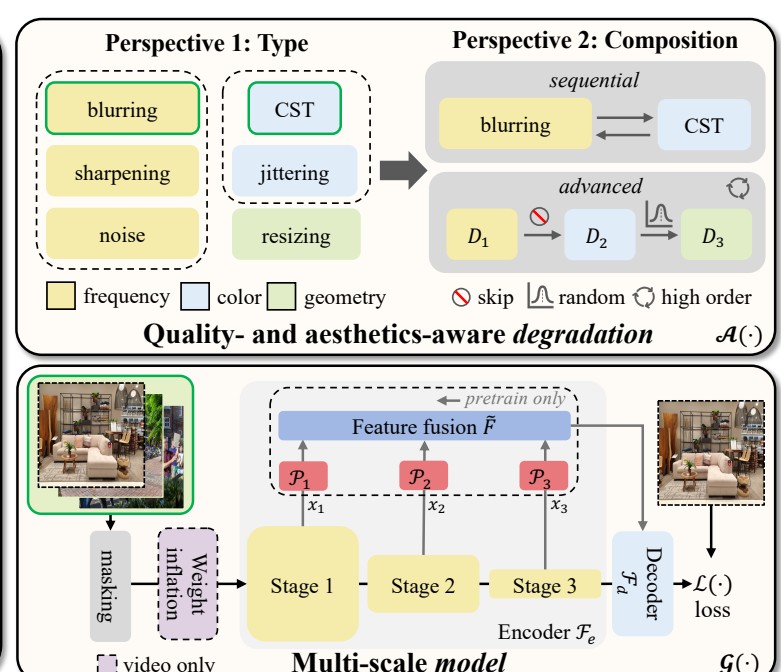

Figure 3: Overview of our proposed QPT V2. QPT V2 incorporates three improvements based on pixel-based MIM tailored for VS. To curate HR & HFC training data, we examine the resolution and foreground coverage of various datasets and samples. To determine quality- and aesthetics-aware degradation, we explore the degradation type and composition. To perceive distortion and aesthetics information in multi-scale fashion, we design a pretrain only feature fusion module based on hierarchical encoder.

## 3 METHODOLOGY

We first revisit MIM concisely in Sec.3.1, and then describe the motivation of QPT V2 in Sec.3.2. Last, the key designs incorporated in QPT V2 are elucidated.

### 3.1 A Revisit of Masked Image Modeling

There are three major steps in MIM: (1) split the image into visible and masked patches, (2) reconstruct the masked patches and (3) calculate the reconstruction loss.

Given the original image $\mathbf{I} \in \mathbb{R}^{H \times W \times 3}$, where $H$, $W$ are the height and width of the image. **First**, specific degradations $\mathcal{A}(\cdot)$ (*e.g.*, resizing) are applied to the image, generating non-overlapping visible patches $\mathbf{I}_v$ and masked patches $\mathbf{I}_m$ with masking $\mathcal{M}$:

$$\mathbf{I}_v = (1 - \mathcal{M}) \odot \mathcal{A}(\mathbf{I})$$
$$\mathbf{I}_m = \mathcal{M} \odot \mathcal{A}(\mathbf{I}) \tag{1}$$

**Second**, only the visible patches $\mathbf{I}_v$ are fed into the autoencoder $\mathcal{G}(\cdot)$ to reconstruct the masked patches $\hat{\mathbf{I}}_m$ as:

$$\hat{\mathbf{I}}_m = \mathcal{G}(\mathbf{I}_v, \mathbf{e}_{[\mathcal{M}]}) \tag{2}$$

The autoencoder $\mathcal{G}(\cdot)$ consists of an encoder $\mathcal{F}_e$ and a decoder $\mathcal{F}_d$, both are stacked Transformer blocks. Here, a shared learnable mask token $\mathbf{e}_{[\mathcal{M}]}$ functions as the placeholder of masked patches, which are combined with the encoder's output and fed into the decoder. **Last**, an MSE loss $\mathcal{L}(\cdot)$ is computed at masked positions for self-supervision as $\mathcal{L} = \|\mathbf{I}_m - \hat{\mathbf{I}}_m\|_2^2$.

### 3.2 Motivation

To accurately score the quality and aesthetics of visual content, a broad range of VS-related factors necessitate examination, namely *high-level* attributes (*e.g.*, semantics, composition *etc.*) and *low-level* distortions (*e.g.*, blur, noise *etc.*). By analysing the insightful features of MIM, we believe the pretrained models have the potential to be both **quality-aware** and **aesthetics-aware**, described next.

**First**, it has been proved that MIM has the ability to comprehend the *high-level semantics* of the image [2, 24]. During pretraining, the large masking ratio forces the model to reconstruct the masked area provided with a few visible patches. Depicted by Fig. 2 (a), the pretrained model gives semantically plausible reconstruction even when 90% of the pixels are masked. **Second**, MIM is proved to be biased towards *low-level details* when reconstructing [51, 52] the *raw pixels*. Due to the perfect reconstruction of pixel values, the model focuses on intricate details (*e.g.*, texture with repeated patterns) besides understanding the content, allowing for a better perception of low-level distortion. To better illustrate the distortion-awareness of MIM, we separately apply blurring and noise to the same image. Reconstruction results in Fig. 2 (b) show that the pretrained model can perceive distortions in low and high frequency intervals, respectively.

Despite MIM has the potential to encompass VS-related factors comprehensively, unleashing its power on downstream VS tasks still presents a non-trivial endeavor. We dissect the MIM framework and identify three components that contribute to this gap:

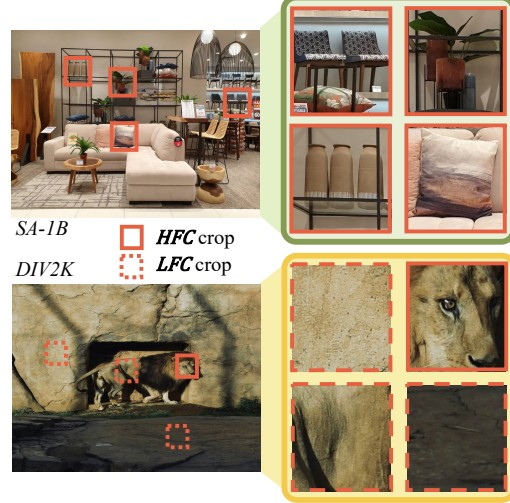

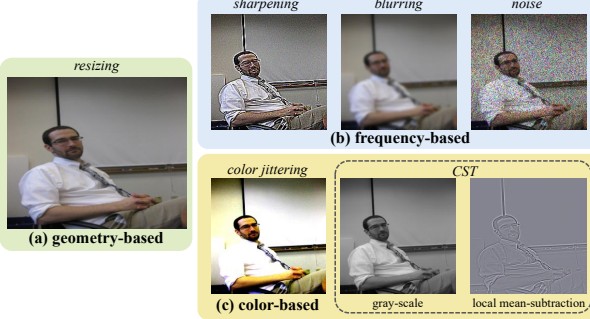

Figure 5: Illustration of the studied degradations, each transforms data stochastically.

**Figure 4: Illustration of the gap in FC between SA-1B and SR datasets after introducing random cropping. Images in SA-1B are more likely to generate HFC crops compared to the images in DIV2K. Please zoom in for a better view.**

- **Data**. ImageNet [11] has become the de-facto pretraining data in MIM studies. The images generally exhibit low resolution and lack intricate details. Over the years, supporting evidence from psychophysical studies has indicated the richness of details (*e.g.*, spatial complexity) in visual content directly impacts human eye's perception of quality and aesthetics [14, 23, 27, 82]. Thus, pretraining on data lacking details might not be sufficient for model to exploit fine-grained quality and aesthetics information. In all, curating **pretraining data** tailored for VS is of utmost importance.
- **Degradation**. MIM achieves excellent results in high-level tasks with simple degradations (*e.g.*, random cropping) [24, 91]. Thus, previous works pay less attention to the degradation design. Yet, simple degradations can only encompass VS-related factors presented in common scenario (*e.g.*, content editing), overlooking other factors introduced by various visual applications, such as compression, transmission, and unprofessional shooting. Therefore, *degradations that cover extensive VS-related factors need to be considered.*
- **Model**. HVS assesses quality and aesthetics in a *multi-scale* fashion [36]. Additionally, numerous previous works have proved the benefits of utilizing multi-scale features in other vision tasks [29, 49]. As a result, to mimic HVS and capture both fine-grained and coarse-grained VS-related factor effectively, the improvement of the **model structure** is exceedingly crucial.

### 3.3 Data

Demonstrated in Fig. 3, pretraining data of QPT V2 is curated from two criteria: **high resolution (HR)** and **high foreground coverage (HFC)**. As argued above, by reconstructing the rich textures and local structures within the HR images, models are prone to perceive a broad range of quality and aesthetics information during pretraining. In addition, FC is defined as the proportion of *foreground region* of the entire image. Since foreground region encodes way more semantics and texture than the background, pretraining on HFC images ensures the model's sensitivity to both high-level and low-level visual attributes.

Based on the two criteria, multiple datasets with various resolution and FC are investigated. We resort to SA-1B [38] as the *pretraining data source* for the following reasons. **First**, SA-1B has an average resolution of approximately 1600×2100, which is significantly higher than that of ImageNet. **Second**, although widely used datasets (*e.g.*, DIV2K [1], UnsplashFull [54] *etc.*) in super-resolution (SR) task possess higher resolution (>2K), SA-1B exhibits a significantly higher FC. To maintain the resolution of HR images while adapting to the small input size of the model (*e.g.*, 224×224), degradations $\mathcal{A}(\cdot)$ in Equ. 1 typically include random cropping, which further widens the gap between SA-1B and SR datasets in terms of FC. Fig. 4 highlights this difference. **Third**, SA-1B provides a straightforward criterion, namely *the number of objects per image*, which allows us to further filter the dataset to get images with higher FC. Eventually, the pretraining dataset for QPT V2 consists of 1.28 million HR images filtered from SA-1B, with each image containing 50 or more objects. The effectiveness of HR & HFC data on downstream VS tasks is validated, we refer the reader to Sec.4.3 for more details.

### 3.4 Degradation

To comprehensively cover the VS-related factors, degradation **type** and **composition** are studied. Fig.5 visualizes all the degradations studied in this work. **First**, to account for VS-related factors introduced by *geometric transformation*, resizing is considered. **Second**, to cover the factors introduced by *frequency shift*, blurring, sharpening, and gaussian noise are studied. **Last**, to incorporate the factors introduced by *color changing*, color jittering and color space transformation (CST) are considered. Following [55], we employ four color spaces including RGB, LAB, HSV and grayscale. Fig.6 showcases the completeness of our degradation selection. In terms of degradation composition, two strategies are adpoted. **First**, we compose degradations *sequentially*. **Second**, inspired by recent progress in SR [85, 111], an *advanced* composition including shuffling, skipping and high-order operations is used to obtain complex degradations. Random cropping is applied after all the degradations by default.

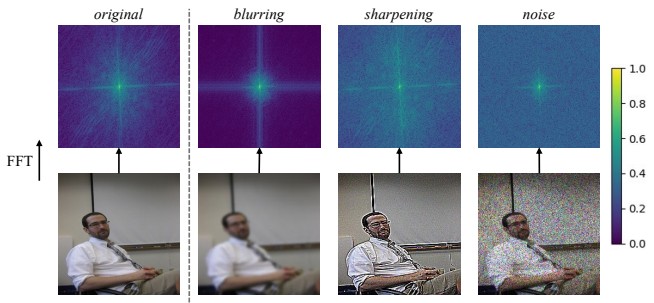

**Figure 6: Illustration of the our comprehensive degradation selection. We only consider frequency-based degradations that result in *different* frequency distributions.**

Benefiting from the comprehensiveness of our degradation selection, we discover that **CST** stands out as the most quality- and aesthetics-aware degradation. Previous NSS-based VS studies have demonstrated that the VS-related information exists in various color spaces [19, 79] and subsequent studies further proposed that the information of different color spaces are *complementary* to each other [55]. Therefore, we speculate that applying CST to the reconstruction target exposes a richer set of quality and aesthetics factors to the model, improving the data diversity during pretraining. Different from previous pretraining objectives based on contrastive learning, we further reveal the fact that QPT V2 does *not* benefit from the sequential or advanced composition of degradations. More details of both findings can be found in Sec.4.3.

### 3.5 Model

To perceive the quality and aesthetics information at different scales, *encoder architecture* and *multi-scale feature fusion* are considered. Regarding the selection of the encoder architecture, the common choices are ViT [13] and hierarchical backbones (*e.g.*, Swin [53] and HiViT [109]). Compared to ViT, hierarchical backbones are better at learning multi-scale features by leveraging image-related inductive biases. Thus, a representative hierarchical backbone HiViT is selected as the encoder.

There are three stages of different scales in HiViT. Upon that, we devise a *fusion module* to incorporate the multi-scale features output by different stages. The fusion process is described next. The hierarchical encoder $\mathcal{F}_e$ outputs features at multiple stages during pretraining, shown in Fig. 3. These features are denoted by $X = \{x_i\}_{1 \le i \le N}$, where $N$ represents the number of stage. **First**, $x_i$ is processed by a *projection layer* $\mathcal{P}_i(\cdot)$, which aligns the feature space between outputs of different stages, as:

$$\overline{X} = \{\mathcal{P}_i(x_i)\}_{1 \le i \le N} \tag{3}$$

**Second**, the projected features of all stages $\overline{X}$, are integrated by a fusion layer $\widetilde{F}(\cdot)$ as:

$$Y = \widetilde{F}(\overline{X}) \tag{4}$$

$Y$ will be fed into the decoder $\mathcal{F}_d$ for pixel reconstruction. Note that the fusion process is only introduced during pretraining, without affecting the finetuning stage. More details of the architecture selection and feature fusion are in Sec.4.3.

## 4 EXPERIMENTS

In this section, experimental setups are first introduced in Sec.4.1. By comparing to existing SOTA methods in Sec.4.2, QPT V2 is evaluated on 11 benchmarks from all three VS tasks. Last, an in-depth ablation over QPT V2 is provided in Sec.4.3.

### 4.1 Evaluation Setups

*Criteria*. SRCC (Spearman rank correlation coef.) and PLCC (Pearson linear correlation coef.) are adopted as evaluation criteria for all three tasks, both ranging in [0, 1]. A larger SRCC indicates a better ranking between samples, and a larger PLCC shows a more accurate score prediction.

*Benchmarks*. 11 benchmarks are selected from IQA, VQA, and IAA to comprehensively evaluate the visual scoring ability of QPT V2. For *IQA*, three synthetically degraded datasets (TID2013 [62], LIVE [68], KADID [48]) and three datasets with real-world distortions (KonIQ10K [32], CLIVE [18], FLIVE [98]) are included. For *VQA*, we choose three public NR-VQA datasets, including LIVE-VQC [71], KoNViD-1k [31], and LSVQ [97]. For *IAA*, AVA [58] is selected for evaluation. The key designs of QPT V2 are ablated on FLIVE, LIVE-VQC, and AVA. For all the datasets without official splitting, we randomly split them into 80% for training and 20% for testing. The finetuning/evaluation procedure is conducted on 10 different splittings to avoid randomness, and the average SRCC and PLCC is reported.

*Pretraining details*. All the experiments are conducted on 4 NVIDIA V100 GPUs. The pretraining data, degradation and model are specified in Sec.3.3, Sec.3.4, and Sec.3.5, respectively. We randomly mask 75% of the pixels following [24] and the input image size is 224×224. The hyperparameter settings are inherited from [24].

*Finetuning strategy*. For *IQA*, we implement the regression head with a simple MLP (*e.g.*, two linear layers with a GeLU activation in between). Following [74], we resize the shorter edge of images to 340 while keeping the aspect ratio, then randomly crop sub-images with size 224×224. AdamW is adopted for optimization, with weight decay of 0.01. The initial learning rate is 2e-5 and decayed by cosine annealing without warmup. Pretrained models are finetuned for 200 epochs, and the checkpoint of the last epoch is selected for evaluation. When testing, we take the four corners and the center crops and average their predicted quality scores to obtain the final score.

For *VQA*, we follow the settings in [89] for finetuning. Also, the pretraining weight is inflated to adapt video input, as done in [75]. As for hyperparamters, AdamW is used with weight decay of 0.05 and mini-batch size of 16. The initial learning rate is set to 1e-3 and decay it with cosine annealing strategy. The pretrained models are finetuned for 30 epochs on LSVQ$_{train}$ followed by evaluating on LSVQ$_{test}$, LSVQ$_{1080p}$ and two other smaller datasets, LIVE-VQC and KoNViD-1k. We uniformly sample four 32-frame clips from an input video, and average the predicted quality scores as the final results.

For *IAA*, pretrained models are finetuned on AVA$_{train}$ for 60 epochs and then evaluate on AVA$_{test}$, images are resized to 224×224

**Table 1: Performance of existing SOTA methods and the proposed QPT V2 on three synthetic and three real-world IQA datasets. "-" means missing corresponding results in the original paper. The best and second-best results are bolded and underlined.**

| Method | Synthetic | | | | | | Real-world | | | | | |
|---|---|---|---|---|---|---|---|---|---|---|---|---|
| | LIVE | | TID2013 | | KADID | | FLIVE | | CLIVE | | KonIQ10K | |
| | SRCC | PLCC | SRCC | PLCC | SRCC | PLCC | SRCC | PLCC | SRCC | PLCC | SRCC | PLCC |
| NIQE [57] | 0.907 | 0.901 | 0.315 | 0.393 | 0.374 | 0.428 | 0.211 | 0.288 | 0.454 | 0.468 | 0.526 | 0.475 |
| BRISQUE [56] | 0.939 | 0.935 | 0.604 | 0.694 | 0.528 | 0.567 | 0.288 | 0.373 | 0.601 | 0.621 | 0.715 | 0.702 |
| ILNIQE [104] | 0.902 | 0.906 | 0.521 | 0.648 | 0.503 | 0.496 | 0.219 | 0.256 | 0.453 | 0.511 | 0.503 | 0.496 |
| CORNIA [96] | 0.947 | 0.950 | 0.678 | 0.768 | 0.516 | 0.558 | - | - | - | - | - | - |
| HOSA [92] | 0.946 | 0.950 | 0.735 | 0.815 | 0.618 | 0.653 | - | - | - | - | - | - |
| DB-CNN [106] | 0.968 | 0.971 | 0.816 | 0.865 | 0.851 | 0.856 | 0.554 | 0.652 | 0.844 | 0.862 | 0.878 | 0.887 |
| HyperIQA [72] | 0.962 | 0.966 | 0.840 | 0.858 | 0.852 | 0.845 | 0.535 | 0.623 | 0.855 | 0.871 | 0.908 | 0.921 |
| CONRTIQUE [55] | 0.960 | 0.961 | 0.843 | 0.857 | **0.934** | **0.937** | 0.580 | 0.641 | 0.854 | 0.890 | 0.896 | 0.901 |
| Re-IQA [66] | 0.970 | 0.971 | 0.804 | 0.861 | 0.872 | 0.885 | 0.645 | **0.733** | 0.840 | 0.854 | 0.914 | 0.923 |
| MUSIQ [36] | - | - | - | - | - | - | 0.566 | 0.661 | - | - | 0.916 | 0.928 |
| TReS [22] | 0.969 | 0.968 | 0.863 | 0.883 | 0.859 | 0.858 | 0.554 | 0.625 | 0.846 | 0.877 | 0.915 | 0.928 |
| QPT [111] | - | - | - | - | - | - | 0.610 | 0.677 | 0.895 | **0.914** | **0.927** | **0.941** |
| QPT V2 | **0.972** | **0.973** | **0.874** | **0.885** | 0.897 | 0.896 | **0.649** | 0.684 | **0.897** | 0.902 | 0.913 | 0.930 |

**Table 2: Performance of existing SOTA methods and the proposed QPT V2 on four in-the-wild VQA datasets. "-" means missing corresponding results in the original paper. The best and second-best results are bolded and underlined.**

| Method | Intra-dataset | | | | Cross-dataset | | | |
|---|---|---|---|---|---|---|---|---|
| | $LSVQ_{test}$ | | $LSVQ_{1080p}$ | | LIVE-VQC | | KoNViD-1k | |
| | SRCC | PLCC | SRCC | PLCC | SRCC | PLCC | SRCC | PLCC |
| BRISQUE [56] | 0.579 | 0.576 | 0.497 | 0.531 | 0.524 | 0.536 | 0.646 | 0.647 |
| TLVQM [39] | 0.772 | 0.774 | 0.589 | 0.616 | 0.670 | 0.691 | 0.732 | 0.724 |
| VIDEVAL [78] | 0.794 | 0.783 | 0.545 | 0.554 | 0.630 | 0.640 | 0.751 | 0.741 |
| VSFA [45] | 0.801 | 0.796 | 0.675 | 0.704 | 0.734 | 0.772 | 0.784 | 0.794 |
| BVQA [44] | 0.852 | 0.854 | 0.772 | 0.788 | 0.816 | 0.824 | 0.839 | 0.830 |
| SimpleVQA [73] | 0.867 | 0.861 | 0.764 | 0.803 | - | - | 0.860 | - |
| $PVQ_{wo/patch}$ [97] | 0.814 | 0.816 | 0.686 | 0.708 | 0.781 | 0.781 | 0.747 | 0.796 |
| $PVQ_{w/patch}$ [97] | 0.827 | 0.828 | 0.711 | 0.739 | 0.770 | 0.807 | 0.791 | 0.795 |
| FastVQA [89] | 0.876 | 0.877 | 0.779 | 0.814 | 0.823 | 0.844 | 0.859 | 0.855 |
| Q-Align [90] | 0.883 | 0.882 | **0.797** | **0.830** | - | - | 0.865 | **0.877** |
| QPT V2 | **0.886** | **0.889** | 0.785 | 0.822 | **0.827** | **0.853** | **0.866** | 0.865 |

for evaluation. The regression head and hyperparameters are kept consistent with those in IQA.

## 4.2 Comparison with state-of-the-arts

*IQA*. We compare our QPT V2 with two groups of IQA methods, including 5 traditional methods and 7 deep learning-based methods. Results in Tab.1 show that QPT V2 achieves superior or comparable performances to current SOTA methods. Previous deep learning-based methods have achieved outstanding performances on three synthetically datasets. *Therefore, further improvements on these datasets can be challenging to attain.* Still, QPT V2 improves the results on LIVE and TID2013 (*e.g.*, **+1.1%** of SRCC on TID2013). Moreover, our method also reaches leading SRCC on FLIVE and CLIVE (+0.4% of SRCC on FLIVE), showcasing its ability to perceive real-world distortions effectively. Besides, Tab.1 includes methods that also harness the power of pretraining by desiging contrastive pretext tasks (*e.g.*, CONRTIQUE, Re-IQA, and QPT). For example,

Re-IQA respectively learns a content-aware encoder on ImageNet-1K and a distortion-aware encoder on 758K distorted images. In comparison, QPT V2 consumes *less* pretraining data and achieves better performance.

*VQA*. We compare QPT V2 to three traditional methods and six deep learning-based methods. Results given in Tab.2 provide the following conclusions. First, the performances we obtain exceed all the traditional methods that rely on hand-crafted features by a large margin, and beat most data-driven methods on four VQA datasets. Second, under the *intra-dataset* setting, QPT V2 pushes the SRCC by 0.3% and PLCC by 0.7% on $LSVQ_{test}$, exhibiting accurate quality assessment. Third, under the *cross-dataset* setting, we surpass the current SOTAs as well (*e.g.*, with 0.4% and 0.9% gains in SRCC and PLCC on LIVE-VQC), presenting impressive generalization capability.

*IAA*. We select 11 deep learning-based methods for comparison. Tab.3 indicates that our method significantly surpasses previous SOTA results on AVA dataset, reaching 0.865 (**+4.3%**) of SRCC and

**Table 3: Performance of existing SOTA methods and the proposed QPT V2 on AVA dataset. The best and second-best results are bolded and underlined.**

| Method | AVA_test SRCC | AVA_test PLCC |
|---|---|---|
| NIMA [15] | 0.612 | 0.636 |
| MLSP [30] | 0.756 | 0.757 |
| AFDC [7] | 0.649 | 0.671 |
| MUSIQ [36] | 0.726 | 0.738 |
| MaxViT [76] | 0.708 | 0.745 |
| CLIP-IQA+ [83] | 0.619 | 0.586 |
| Aesthetic Predictor [42] | 0.721 | 0.723 |
| TANet [28] | 0.758 | 0.765 |
| GAT$_{\times 3}$-GATP [21] | 0.762 | 0.764 |
| LIQE [108] | 0.776 | 0.763 |
| VILA [37] | 0.774 | 0.774 |
| Q-Align [90] | 0.822 | 0.817 |
| QPT V2 (60% finetuning data) | 0.766 | 0.780 |
| QPT V2 | **0.865** | **0.875** |

**Table 4: Comparisons of end-to-end finetuning evaluation using different pretext tasks on CLIVE and LIVE-VQC, and AVA.**

| Pretext task | CLIVE SRCC | CLIVE PLCC | LIVE-VQC SRCC | LIVE-VQC PLCC | AVA SRCC | AVA PLCC |
|---|---|---|---|---|---|---|
| QPT V2 | 0.645 | 0.684 | 0.827 | 0.853 | 0.865 | 0.875 |
| QPT [111] | 0.610 | 0.677 | - | - | - | - |
| MoCo [25] | 0.578 | 0.629 | 0.819 | 0.828 | 0.707 | 0.712 |
| Supervised | 0.556 | 0.604 | 0.810 | 0.825 | 0.704 | 0.690 |
| w/o | 0.451 | 0.475 | 0.696 | 0.731 | 0.545 | 0.552 |

0.875 (**+5.8%**) of PLCC. It is worth noting that Q-Align [90] leverages the power of large multi-modality models (LMMs). In comparison, our work introduces a new pretraining paradigm, and exhibits lower computation and smaller model size. The advantages become more evident when comparing to methods *without utilizing LMMs* (*e.g.*, **+8.9%** of SRCC and **+10.1%** of PLCC). The finetuning data amount is further reduced to investigate the power of QPT V2. The results show that QPT V2 achieves parity with some previous SOTA methods (*e.g.*, LIQE, VILA) using only **60%** finetuning data, realizing a more data-efficient transfer. Both LIQE and VILA solve IAA by using auxiliary knowledge in text description. In comparison, QPT V2 achieves SOTA results without the assistance of text modality.

**QPT V2 vs. other pretext tasks**. We compare QPT V2 with four pretext tasks, including QPT, MoCo, ImageNet-1K supervised and train-from-scratch, on three representative VS benchmarks. Note that both supervised training and train-from-scratch use the same encoder backbone as QPT V2, which is HiViT. MoCo and QPT are based on semantic-aware and quality-aware contrastive learning, respectively. The results in Tab.4 verify the superiority of QPT V2. In addition, QPT V2 also achieves better performances than the supervised training and the one without pretrained weights in all three VS tasks.

## 4.3 Ablation Studies

**Table 5: Ablation on resolution and foreground coverage of pretraining data. IN1K and UF denote ImageNet-1K and UnsplashFull for simplicity.**

| Source | HR | HFC | FLIVE SRCC | FLIVE PLCC | LIVE-VQC SRCC | LIVE-VQC PLCC | AVA SRCC | AVA PLCC |
|---|---|---|---|---|---|---|---|---|
| IN1K | ✗ | ✔ | 0.617 | 0.653 | 0.812 | 0.825 | 0.759 | 0.780 |
| UF | ✔ | ✗ | 0.602 | 0.631 | 0.799 | 0.828 | 0.778 | 0.801 |
| SA-1B | ✔ | ✔ | 0.645 | 0.684 | 0.827 | 0.853 | 0.865 | 0.875 |

**Table 6: Ablation on single degradation type, each transforms data stochastically.**

| Deg. | FLIVE SRCC | FLIVE PLCC | LIVE-VQC SRCC | LIVE-VQC PLCC | AVA SRCC | AVA PLCC |
|---|---|---|---|---|---|---|
| None | 0.616 | 0.664 | 0.813 | 0.836 | 0.832 | 0.820 |
| Resizing | 0.593 | 0.621 | 0.797 | 0.815 | 0.774 | 0.752 |
| Blurring | 0.628 | 0.664 | 0.813 | 0.833 | 0.827 | 0.831 |
| Sharpening | 0.617 | 0.650 | 0.803 | 0.820 | 0.801 | 0.786 |
| Noise | 0.602 | 0.614 | 0.793 | 0.810 | 0.773 | 0.747 |
| CST | 0.645 | 0.684 | 0.827 | 0.853 | 0.865 | 0.875 |
| Color jittering | 0.623 | 0.649 | 0.809 | 0.826 | 0.788 | 0.792 |

**Table 7: Ablation on different forms of degradation composition. CST and B denote color space transform and blurring for simplicity.**

| Comp. | Deg. | FLIVE SRCC | FLIVE PLCC | LIVE-VQC SRCC | LIVE-VQC PLCC | AVA SRCC | AVA PLCC |
|---|---|---|---|---|---|---|---|
| None | CST | 0.645 | 0.684 | 0.827 | 0.853 | 0.865 | 0.875 |
| None | B | 0.628 | 0.664 | 0.813 | 0.833 | 0.827 | 0.831 |
| Sequential | B→CST | 0.645 | 0.671 | 0.806 | 0.824 | 0.821 | 0.840 |
| Sequential | CST→B | 0.637 | 0.674 | 0.815 | 0.838 | 0.855 | 0.874 |
| Advanced | All | 0.603 | 0.652 | 0.797 | 0.811 | 0.820 | 0.839 |

**Effectiveness of HR & HFC data**. We demonstrate the effectiveness of HR & HFC data in QPT V2 by comparing to models pretrained on data with differenet resolution and FC. Following conclusions can be drawn from Tab.5. **First**, pretraining on data with both HR and HFC leads to best downstream performances. **Second**, high resolution matters. When the foreground coverage is generally high, pretraining on HR images yields noticeably superior performance on *all three* representative VS datasets. *e.g.*, +2.9% on FLIVE, +2.2% on LIVE-VQC and +10.1% on AVA. **Last**, HFC is essential. When the resolution is relatively high, HFC data always prevail. *e.g.*, +4.8% on FLIVE, +2.7% on LIVE-VQC, and +8.1% on AVA.

**Effectiveness of quality- and aesthetics-aware degradation**. Tab.6 displays the downstream performances after applying six different degradations to the reconstructed target in QPT V2. Results obtained without employing any form of degradation serve as the baseline. **First**, the CST degradation incorporated in QPT V2 performs the best, demonstrating its quality- and aesthectics-awareness. **Second**, blurring brings a slight improvement in IAA (+0.5% of SRCC and +1.1% of PLCC). Recent MIM studies find that

removing the high-frequency components of pixels helps the model to focus on semantics, benefiting downstream high-level tasks [51]. Thus, we attribute the gains to the fact that IAA places greater emphasis on high-level visual attributes compared to IQA and VQA [90]. **Last**, the geometry-based resizing, frequency-based sharpening and noise, and color jittering impair the downstream performances on all three benchmarks. This suggests that, altering the spatial layout of the reconstruction target, enriching its high-frequency details, corrupting its frequency spectrum, or perturbing its color are all detrimental to model's quality- and aesthetics-awareness.

To study the effect of degradation composition, two top-performing degradations in Tab.6, namely, CST and blurring are arranged sequentially. Also, all six degradations are included in an advanced composition protocol discussed in Sec.3.4. The results in Tab.7 indicate two findings. **First**, CST and blurring cannot synergize when arranged sequentially, which leads to slight inferior results. **Second**, QPT V2 does not benefit from a complicated degradation space. Above findings are inconsistent with those in contrastive learning, we attribute them to the distinction between two pretraining paradigms.

***Effectiveness of multi-scale model***. Tab.8 validates the effectiveness of the *encoder architecture selection*. With similar model capacity, hierarchical backbones outperform plain ViT in *all three* VS tasks. For example, Swin-T reaches 0.868 of SRCC on AVA dataset, 4.2% higher than ViT-S. Since two hierarchical models exhibit similar downstream performances, we opt for HiViT-T as the encoder in QPT V2 for better training efficiency. Though increasing the model capacity might potentially yield better results, we did not use larger models out of tradeoffs, which can be done in future work.

To validate the effectiveness of the multi-scale feature fusion strategy proposed in Sec.3.5, we fuse features at different stages, and the downstream results are given in Tab.9. By default, the output of the last stage (stage 3) is always fed to the decoder. Tab.9 indicates that multi-scale feature fusion always provides benefits for VS tasks. Particularly, fusing features from shallow stage (stage 1) yields the most significant gains on three downstream datasets. Due to the inclusion of more low-level details in shallow layer features, we believe that fusing these features assist the model in better perceiving low-level VS-related factors. Additionally, the implementation choices of the projection layer $\mathcal{P}_i(\cdot)$ and the fusion layer $\widetilde{F}(\cdot)$ specified in Sec.3.5 are discussed in Tab.10. Following conclusions can be drawn: **First**, a simple linear layer is sufficient to project representation into the same feature space. A more complex MLP (*e.g.*, Linear-GeLU-Linear structure) cannot bring improvement while introducing non-negligible computational overhead. We think the non-linearity may increase the optimization difficulty as for pretraining. **Second**, weighted average pooling is better suited for integrating projected features compared to the simple summation.

***Impact of pretraining data amount***. We study the impact of data amount on QPT V2 by using 20%, 50% and 100% percentages of the pretraining data. Given by Tab.11, the performances on three downstream datasets continue to improve as the data amount increases. Surprisingly, we find that even when using only **50%** of the data, QPT V2 still achieves comparable performances to SOTA

**Table 8: Ablation on the selection of the encoder architecture. MS denotes to multi-scale for simplicity.**

| Model | MS | Param | FLIVE | | LIVE-VQC | | AVA | |
|---|---|---|---|---|---|---|---|---|
| | | | SRCC | PLCC | SRCC | PLCC | SRCC | PLCC |
| ViT-S | ✗ | 22M | 0.614 | 0.651 | 0.809 | 0.835 | 0.822 | 0.832 |
| Swin-T | ✔ | 28M | 0.647 | 0.671 | 0.828 | 0.850 | 0.868 | 0.863 |
| HiViT-T | ✔ | 19M | 0.645 | 0.684 | 0.827 | 0.853 | 0.865 | 0.875 |

**Table 9: Ablation on the location for feature fusion.**

| Stage 1 | 2 | FLIVE | | LIVE-VQC | | AVA | |
|---|---|---|---|---|---|---|---|
| | | SRCC | PLCC | SRCC | PLCC | SRCC | PLCC |
| ✗ | ✗ | 0.643 | 0.650 | 0.814 | 0.837 | 0.848 | 0.858 |
| ✔ | ✗ | 0.645 | 0.684 | 0.827 | 0.853 | 0.865 | 0.875 |
| ✗ | ✔ | 0.643 | 0.671 | 0.819 | 0.838 | 0.842 | 0.863 |
| ✔ | ✔ | 0.654 | 0.672 | 0.818 | 0.838 | 0.854 | 0.869 |

**Table 10: Ablation on different implementatons of the multi-scale feature fusion. Linear and MLP represents linear and MLP projection layer, while Pool and Sum denote the fusion strategies of weighted-average pooling and summation.**

| Linear | MLP | Pool | Sum | FLIVE | | LIVE-VQC | | AVA | |
|---|---|---|---|---|---|---|---|---|---|
| | | | | SRCC | PLCC | SRCC | PLCC | SRCC | PLCC |
| ✔ | | ✔ | | 0.645 | 0.684 | 0.827 | 0.853 | 0.865 | 0.875 |
| ✔ | | | ✔ | 0.638 | 0.667 | 0.804 | 0.821 | 0.820 | 0.830 |
| | ✔ | ✔ | | 0.656 | 0.682 | 0.820 | 0.848 | 0.858 | 0.864 |

**Table 11: Impact of data amount for the pretext task of QPT V2, using different percentages of the pretraining dataset.**

| Percentage | FLIVE | | LIVE-VQC | | AVA | |
|---|---|---|---|---|---|---|
| | SRCC | PLCC | SRCC | PLCC | SRCC | PLCC |
| 20% | 0.529 | 0.533 | 0.772 | 0.796 | 0.586 | 0.602 |
| 50% | 0.610 | 0.647 | 0.805 | 0.829 | 0.770 | 0.774 |
| 100% | 0.645 | 0.684 | 0.827 | 0.853 | 0.865 | 0.875 |

methods in IAA (*e.g.*, 0.770 of SRCC, 0.774 of PLCC on AVA dataset), demonstrating its strong aesthetics perception capabilities. Furthermore, using different data sources to organize more HR & HFC data remains an open question and could be explored in future work.

## 5 CONCLUSION

We propose QPT V2, a novel MIM-based pretraining paradigm crafted for visual scoring tasks, aiming at alleviating the obstacle of insufficient annotated data. To enhance the quality- and aesthetics-awareness of the pretraining objective, we provide a meticulous analysis over the vanilla MIM framework and make targeted improvements on three key components: pretraining data, degradation, and model structure. After pretraining, QPT V2 achieves SOTA results on 11 downstream benchmarks in IQA, VQA, and IAA, demonstrating impressive capability and generalization ability. In all, we hope this work will inspire the community to reflect and explore the possibility of different pretraining paradigms in the context of visual scoring.

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
