# OpenReview forum: "QPT-V2: Masked Image Modeling Advances Visual Scoring"
_acmmm.org/ACMMM/2024/Conference — MM2024 Poster_

### Official Review · Reviewer_wcjm · 2024-05-16

**Rating:** 1
**Confidence:** 4

**Summary:**

According to the Call for Submission(https://2024.acmmm.org/regular-papers)
"Submitted papers may consist of up to 8 pages. Up to two additional pages may be added for references. The reference pages must only contain references. Overlength papers will be rejected without review."

The references pages are more than two pages.

**Strengths:**

None

**Suitability:**

3

---

### Official Review · Reviewer_HXT3 · 2024-05-24

**Rating:** 4
**Confidence:** 2

**Summary:**

This paper proposes a new MIM method towards image quality and aesthetics. The resulting model can be fine-tuned on various visual scoring benchmarks and achieve SOTA results.

**Strengths:**

1. The paper is well-writen and the motivation is clear.
2. The proposed solutions towards data, degradation and model are reasonable.
3. The Experimental results are strong.

**Limitations:**

The analysis of why different degradation approaches result in different visual scoring results is unclear. A deeper explanation and analysis of the correspondence between degradation and model’s quality- and aesthectics-awareness could make the paper more acceptable.

**Suitability:**

3

---

### Official Review · Reviewer_hx8V · 2024-05-24

**Rating:** 6
**Confidence:** 3

**Summary:**

This work aims at learning quality- and aesthetics-aware representation via Mask Image Modeling and further finetune the encoder on various downstream tasks. The designed scheme showcases its reasonable effectiveness on performance improvements and sufficient ablation studies consolidate conclusions of each modules.

**Strengths:**

1. This paper has its clear research objection that is to learn quality and aesthetics-aware representation, which is pivotal in IAA and IQA.

2. Instead of designing modules with huge parameters or large-scale inputs, they think and utilize representation techniques and adapted it into IQA research field, leading a good technical novelty and SOTA performance.

3. This work has clear writing structure and good experiment design. It indicates that the authors have their own thinking and opinions clearly.

**Limitations:**

1. In MIM, it would be great if the author could have discussions on downstream performance change caused by different mask ratios in the pretraining stage. According to line 553, the author claim that they randomly mask 75% by following prior work which is fine, but would be great to have this in hyper-param study.

2. In section 3.3, the authors claim that QPT-v2 is curated from two criteria, i.e. HR and HFC and also provide reasons of choosing SA-1B as pretraining data source. It would be great if authors can provide any quantitative statistic results here for proving the importance of HR and HFC. According to line 433, saying SA-1B exhibits a significantly higher FC compared with SR task databases. And indeed, I also see table 5 in the ablation study section, which can prove the related conclusions.

**Suitability:**

3

---

### Official Review · Reviewer_obdk · 2024-05-29

**Rating:** 4
**Confidence:** 3

**Summary:**

This submission proposes a novel pretraining framework based on Masked Image Modeling (MIM) designed to improve performance on Visual Scoring (VS) tasks. The framework unifies the tasks of Image Quality Assessment (IQA), Visual Quality Assessment (VQA), and Image Aesthetics Assessment (IAA) under a single methodology. The proposed model incorporates high-resolution and high-foreground coverage data, quality and aesthetics-aware degradations, and a multi-scale model structure.

**Strengths:**

1) The extensive experiments across 11 benchmarks provide strong evidence of the effectiveness of the proposed method.
2) The paper is well-organized, and the method are clearly explained, which makes the proposed framework easy to follow.
3) The use of MIM in the context of VS tasks is a novel idea. The decomposition of MIM into data, degradation, and model components is well-thought-out.

**Limitations:**

1) Although there are some ablation studies, more detailed experiments isolating the impact of each component (data curation, degradation types, multi-scale modeling) would strengthen the claims about their contributions.
2) As a conference paper, the number of references is too much and it is advisable to organize them properly.

**Suitability:**

3

---

### Meta-Review · Area_Chair_y5dr · 2024-07-11

**Recommendation:** Accept (Poster)
**Confidence:** 3

**Metareview:**

This submission gets final ratings of 1 accept, 2 weak accept and 1 borderline accept.
It could be accepted as a poster.